# An amino acid substitution in HCV core antigen limits its use as a reliable measure of HCV infection compared with HCV RNA

**Payuda Hansoongnern**[1], **Pornpitra Pratedrat**[1], **Pornjarim Nilyanimit**[1], **Rujipat Wasitthankasem**[1,2], **Nawarat Posuwan**[1], **Nasamon Wanlapakorn**[1], **Kanchanok Kodchakorn**[iD][3], **Prachya Kongtawelert**[3], **Napaporn Pimsing**[4], **Yong Poovorawan**[iD][1,5]*

**1** Center of Excellence in Clinical Virology, Department of Pediatrics, Faculty of Medicine, Chulalongkorn University, Bangkok, Thailand, **2** National Biobank of Thailand, National Science and Technology Development Agency, Pathum Thani, Thailand, **3** Thailand Excellence Center for Tissue Engineering and Stem Cells, Department of Biochemistry, Faculty of Medicine, Chiang Mai University, Chiang Mai, Thailand, **4** Non-Communicable Disease Control Group, Phetchabun Provincial Health Office, Phetchabun, Thailand, **5** Fellow of the Royal Society of Thailand (FRS[T]), the Royal Society of Thailand, Sanam Sueapa, Bangkok, Thailand

* yong.p@chula.ac.th

**Data Availability Statement:** HCV core nucleotide sequences from this study are available from GenBank (https://www.ncbi.nlm.nih.gov/genbank/)

## Abstract

Hepatitis C virus (HCV) is a viral pathogen that causes chronic hepatitis, which can lead to cirrhosis and hepatocellular carcinoma. Detection of HCV RNA is the standard method used to diagnose the disease and monitor antiviral treatment. A quantification assay for the HCV core antigen (HCVcAg) has been proposed as a simplified alternative to the HCV RNA test for predicting active HCV infection, with the aim of achieving the global goal of eliminating hepatitis. The objective of this study was to determine the correlation between HCV RNA and HCVcAg, as well as the impact of amino acid sequence heterogeneity on HCVcAg quantification. Our findings demonstrated a strong positive correlation between HCV RNA and HCVcAg across all HCV genotypes (1a, 1b, 3a, and 6), with correlation coefficients ranging from 0.88 to 0.96 ($p < 0.001$). However, in some cases, samples with genotypes 3a and 6 exhibited lower HCVcAg levels than expected based on the corresponding HCV RNA values. Upon the core amino acid sequence alignment, it was observed that samples exhibiting low core antigen levels had an amino acid substitution at position 49, where threonine was replaced by either alanine or valine. Core mutation at this position may correlate with one of the epitope regions recognized by anti-HCV monoclonal antibodies. The present findings suggest that the utilization of HCVcAg as a standalone marker for HCV RNA might not provide adequate sensitivity for the detection of HCV infection, especially in cases where there are variations in the amino acid sequence of the core region and a low viral load of HCV RNA.

under the accession numbers OQ351363-
OQ351716.

**Funding:** "This study was part of a viral hepatitis elimination project in Thailand and was supported by the National Research Council of Thailand, Thailand Grand Challenge Fund (RES_64_058_30_020), Center of Excellence in Clinical Virology, Faculty of Medicine, Chulalongkorn University, and King Chulalongkorn Memorial Hospital. The funders had no role in study design, data collection and analysis, decision to publish, or preparation of the manuscript"

**Competing interests:** The authors have declared that no competing interests exist.

## Introduction

The hepatitis C virus (HCV) is a causative agent of acute and chronic hepatitis worldwide. The World Health Organization (WHO) estimates that about 58 million people have chronic HCV infection, and 1.5 million new cases of HCV infection occur each year [1]. Approximately 70% of HCV-infected individuals develop chronic HCV infection, which places them at risk for hepatic fibrosis, cirrhosis, and hepatocellular carcinoma [1, 2]. Although there is no vaccine for HCV, direct-acting antivirals (DAAs) have revolutionized treatment, achieving cure rates of over 95% [3]. The World Health Organization (WHO) has launched a global strategy to eliminate viral hepatitis by 2030, which aims to increase screening and treatment rates, reduce new infections, and ultimately reduce hepatitis-related mortality by 60% [4].

Over the past decade, Thailand has observed a decline in HCV infection rates [5, 6]. However, approximately 1% of the general population (around 700,000 people) still carry HCV antibodies, and about 400,000 are viremic [6]. To support the global hepatitis elimination strategy, the Ministry of Public Health (MoPH) in Thailand has integrated HCV diagnostic screening and treatment cost subsidization into the Universal Health Coverage program for the general population [7]. The standard diagnosis for HCV infection involves testing for anti-HCV antibodies and confirming the presence of chronic infection by testing for HCV RNA in antibody-positive individuals [1, 8]. HCV RNA detection is considered the gold standard for diagnosing HCV infection and monitoring treatment due to its high sensitivity and specificity [9]. HCV core antigen (HCVcAg) quantification also correlates well with HCV RNA and HCVcAg levels [10–12] and is suggested as an alternative approach to evaluate active HCV infection and monitor the success of antiviral therapy [9, 13, 14]. HCVcAg is cost-effective and uses the same testing platform as HCV antibody tests, making it useful for HCV diagnostic screening. However, its effectiveness in monitoring treatment response is debatable due to limitations in detecting low viral titers [15]. To address this concern, the Thai National Health Security Office (NHSO), MoPH has implemented HCVcAg and HCV RNA assays as confirmatory diagnostic tests for the 2023 HCV reimbursement program. In addition, the use of HCVcAg as an alternative marker to HCV RNA for population diagnostic screening and confirmatory HCV infection needs to be evaluated.

HCVcAg is a 21-kDa structural protein consisting of 191 amino acids [16] that are highly conserved among different HCV genotypes [17]. Genetic variability in the core protein can occur, however, posing a challenge for accurately detecting HCV infection. A prior study described an association between amino acid substitutions at positions 48 (A48T) and 49 (T49A/P) in the HCV core region and false negative HCVcAg results [18]. Thus, the current study sought to assess the relationship between HCV RNA and core antigen levels by HCV genotype and to determine the effect of amino acid sequence heterogeneity on HCVcAg quantification.

## Materials and methods

### Sample collection

A total of 354 serum samples were collected from patients in Phetchabun province, which has a high prevalence of HCV infection compared to other areas in Thailand. The samples were obtained as leftover plasma from a previous diagnostic screening project that included HCV antibody testing, as well as qualitative and quantitative HCV RNA testing (viral load), as described in a previous study [19]. Briefly, all samples were screened for anti-HCV antibodies with a rapid diagnostic test (Bioline HCV, Abbott Diagnostics, Korea Inc. Korea). Samples initially positive for anti-HCV were subjected to a qualitative HCV RNA assay by using the

COBAS AmpliPrep/COBAS TaqMan HCV Test, v2.0 (Roche Molecular Systems, Pleasanton, CA). The study was approved by the Institutional Review Board of the Faculty of Medicine of Chulalongkorn University (IRB Number 028/63) and was conducted under the principles of the Declaration of Helsinki. Written informed consent was obtained from participants before enrollment.

## HCV genotyping

Viral RNA was extracted from sera by using the QIAamp Viral RNA Mini Kit (Qiagen, Hilden, Germany) according to the manufacturer's instructions, after cDNA synthesis using ImProm-II Reverse Transcription System (Promega, Madison, WI). Of the 354 HCV RNA-positive samples, partial core amplification was conducted in two groups. The initial group of 76 samples was genotyped using a previously published primer set [20], resulting in nucleotide sequences that were insufficiently long for analysis of the three-dimensional modeling of amino acid substitutions in the core region or re-amplification due to sample depletion. Therefore, the remaining 278 samples were subjected to the amplification of the core protein gene using a new primer set, forward primer HCV252F (5' TAGCCGAGTAGTGTTGGGGTC 3') and reverse primer HCV410R (5' ATGTACCCCATGAGGTCGGC 3'). The forward and reverse primers bound the 5'UTR at nucleotide positions 251–270 and the core region at nucleotide positions 732–751, respectively (based on H77, HCV reference strain with the accession number NC_004102). The partial core gene was amplified using AccuStart II Gel-Track PCR Super Mix (QuantaBio, Beverly, MA). The PCR conditions included an initial denaturation at 95˚C for 3 minutes followed by 40 cycles of denaturation at 95˚C for 30 seconds, annealing at 48˚C for 30 seconds, and extension at 72˚C for 45 seconds, with a final extension at 72˚C for 7 minutes. Amplicons were purified using GeneAll Expin™ Gel SV (GeanAll Biotechnology CO., LTD., Seoul, Korea) and subjected to Sanger sequencing (1st Base, Malaysia). HCV genotypes were classified by phylogenetic analysis after alignment with reference sequences from each HCV genotype available from the GenBank database (mentioned below). HCV core nucleotide sequences from this study were submitted to GenBank under the accession numbers OQ351363-OQ351716.

Multiple sequence alignments were generated using Clustal X version 2.0 [21], with the reference genome as follows; 1a_M62321, 1a_M67463, 1b_D90208, 1b_M58335, 1c_D14853, 1g_AM910652, 1h_KC248198, 2a_D00944, 2a_AB047639, 2b_AB030907, 2c_D50409, 3a_D17763, 3a_D28917, 3b_D49374, 4a_Y11604, 5a_AF064490, 5a_Y13184, 6a_Y12083, 6a_AY859526, 6b_D84262, 6c_EF424629, 6d_D84263, 6e_DQ314805, 6f_DQ835760, 6g_D63822, 6h_D84265, 6i_DQ835770, 6j_DQ835769, 6k_D84264, 6l_EF424628, 6m_DQ835767, 6n_DQ278894, 6n_DQ835768, 6o_EF424627, 6p_EF424626, 6q_EF424625, 6r_EU408328, 6s_EU408329, 6t_EF632071, 6t_EU246939, 6u_EU246940, 6v_EU158186, 6v_EU798760, 6w_DQ278892, 6w_EU643834, 6xa_EU408330, 7a_EF108306. Phylogenetic trees were constructed using the Neighbor-joining approach with Kimura's 2 parameters model. Bootstrap resampling tree were generated with 1000 replicates. HCV genotype was assigned according to the same cluster with the respective reference genotype (S1–S3 Figs).

## HCV RNA and HCVcAg quantification

HCV RNA viral load was quantified by RT-PCR using the cobas 4800 System (Roche Diagnostics, Manheim, Germany) according to the manufacturer's instructions. The upper and lower limits of detection for HCV genotypes 1–6 were $10^8$ and 15 IU/ml, respectively. HCVcAg was evaluated by Abbott Architect i1000SR analyzer using the ARCHITECT HCV Ag assay (Abbott Diagnostics, Wiesbaden, Germany), a quantitative chemiluminescent immunoassay

utilizing the anti-HCV monoclonal antibody-coated microparticles to detect HCVcAg. Briefly, the sample is first combined with pre-treatment 1 and 2 reagents before being mixed with assay-specific diluents and anti-HCV-coated microparticles. After washing, acridinium-labeled anti-HCV conjugate is added to the mixture, and then pre-trigger and trigger solutions are added. The resulting reaction is measured as relative light units (RLUs). The concentration of HCVcAg in the sample is determined by comparing the RLUs to a previously generated ARCHITECT HCV Ag calibration curve. Sample values $\geq$ 3 fmol/L ($\geq$ 0.477 Log10 fmol/L) and < 3 fmol/L were considered positive and non-reactive, respectively. Samples with concentration values ranging from $\geq$ 3.00 fmol/L to < 10.00 fmol/L underwent duplicate testing, and those that were retested with a concentration $\geq$ 3.00 fmol/L were considered reactive positive. The Abbott ARCHITECT HCV Ag assay, developed by Abbott HmbH & Co. KG in Wiesbaden, Germany, demonstrated a sensitivity of 97.8% and specificity greater than 99.5%.

## Statistical analysis

Data were log10 transformed and analyzed using Spearman's rank correlation and Mann-Whitney U tests with IBM SPSS statistics software (IBM Corporation, Armonk, NY). A $p$-value of < 0.05 was considered significant.

## Results

All 354 HCV antibody-positive samples possessed detectable HCV RNA. HCV genotyping on these samples revealed genotype 1a (n = 56, 15.8%), 1b (n = 23, 6.5%), 3a (n = 107, 30.2%), and 6 (n = 168, 47.5%) (Table 1). HCVcAg was detectable in the majority of the samples (> 95%, mean = 5,699 fmol/L) and showed that levels of HCVcAg somewhat correlate with HCV RNA (Fig 1). Two genotype 6 samples had HCV RNA levels of 4.04 x 10³ and 8.85 x 10³ IU/ml but were non-reactive with HCVcAg values < 3 fmol/L. This positive correlation between HCV RNA and HCVcAg was found in samples from all genotypes ($p$ < 0.001). The highest and lowest correlation was observed for samples of genotype 1b (r = 0.96) and genotype 6 (r = 0.88), respectively. For many samples of genotypes 3a and 6, however, HCVcAg levels were lower than would be expected.

There was no significant difference in the HCVcAg/HCV RNA ratio for samples of genotype 1a, 1b, and 6. In contrast, samples of genotype 3a had a lower HCVcAg/HCV RNA ratio than the genotype 1a, 1b, and 6 samples ($p$ < 0.05), which suggests that the levels of HCVcAg were lower relative to HCV RNA for genotype 3a samples (Fig 1). Although certain genotype 6 samples had lower levels of HCVcAg in relation to HCV RNA, the HCVcAg/HCV RNA ratio did not exhibit a significant difference between samples with genotype 6 and genotype 1. Nevertheless, slight variation was observed in the HCVcAg/HCV RNA ratios among genotype 6 samples, which were subsequently analyzed for amino acid substitutions.

**Table 1. Amino acid residue in the HCVcAg at position 49 for each genotype.**

| Genotype | No. of samples (%) | Average HCV RNA level (IU/ml) | The amino acid at residue 49, No. for each genotype (%) | |
|---|---|---|---|---|
| | | | T | A or V |
| 1a | 56 (15.8%) | 5.01 x 10⁶ | 56 (100%) | - |
| 1b | 23 (6.5%) | 4.09 x 10⁶ | 23 (100%) | - |
| 3a | 107 (30.2%) | 4.70 x 10⁶ | 100 (93.5%) | 7 (6.5%) |
| 6 | 168 (47.5%) | 8.02 x 10⁶ | 162 (96.4%) | 6 (3.6%) |
| Total | 354 (100%) | | | |

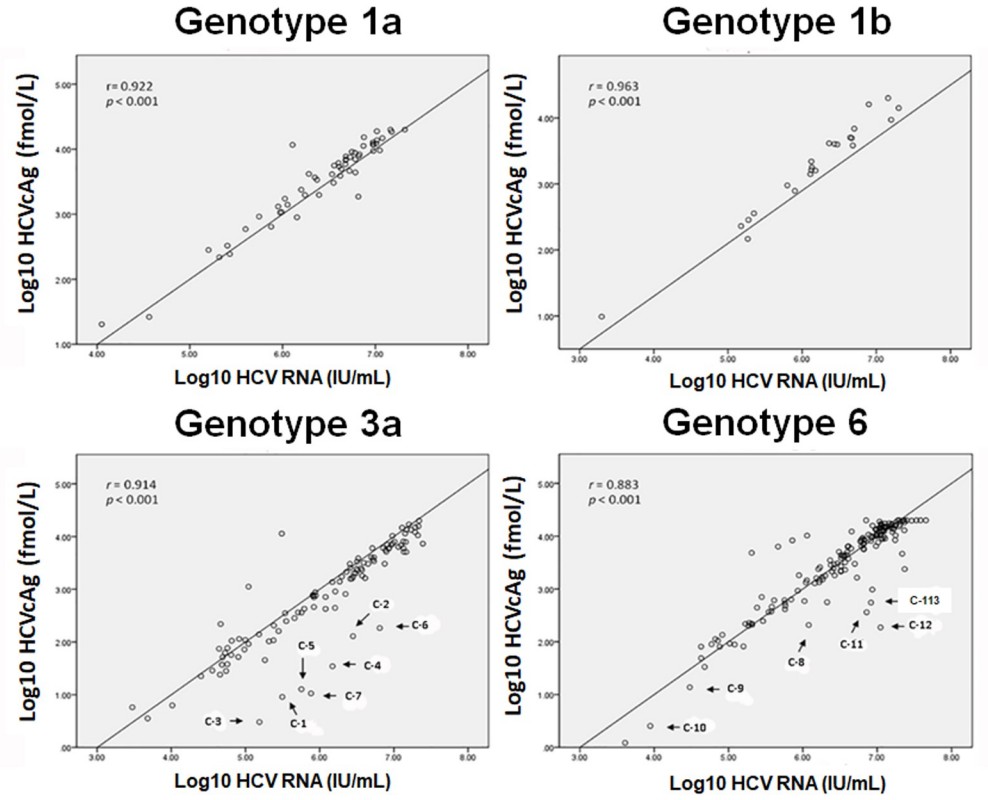

**Fig 1. Correlations between HCV RNA and HCVcAg for different HCV genotypes.** HCV RNA viral loads in IU/ml (x-axis) were plotted against detectable HCVcAg in fmol/L (y-axis) for samples (circles). Correlation scores from the line plot (r-value) and statistical significance (p-value) are indicated for each HCV genotype. Samples with lower levels of HCVcAg relative to the HCV RNA are arrowed for genotype 3a (samples C-1 to C-7) and genotype 6 (samples C-8 to C-13).

Samples associated with lower HCVcAg levels than expected from their corresponding HCV RNA values were classified as genotypes 3a and 6. Amino acid sequences of the samples with low HCVcAg levels (samples C-1 to C-13) were compared with randomly selected samples with core antigen levels corresponding to HCV RNA values (samples N-1 to N-27) to determine the amino acid substitution associated with the HCVcAg underestimate. While the C-1 to C-7 and N-1 to N-15 were characterized as genotype 3a, the C-8 to C-13 and N-16 to N-27 were characterized as genotype 6. The accession number of case (C) and control (N) samples is shown in supplementary data (S1 File).

The amino acid sequence alignment of partial HCVcAg residues of C-1 to C-13 and N-1 to N-27 showed high conservation (Fig 2). Thirteen samples (C-1 to C-13) from both genotypes 3a and 6 had a core mutation at position 49 (threonine to alanine or valine) compared to control samples that lacked an amino acid substitution. These findings suggest that a core mutation at residue 49 may affect the levels of HCVcAg determined using the ARCHITECT HCV Ag platform.

The tertiary structure of the control and mutant (T49A/V) HCV core protein was predicted using the IntFOLD server. According to the server's quality and confidence scoring, the global model quality scores of the tertiary structure ranged from 0 to 1. Scores > 0.4 had more complete and confident models and were very close to the native structure. Peptides 2–45 of the HCV N-terminal core protein (PDB ID: 1CWX) were compared with the predicted HCV

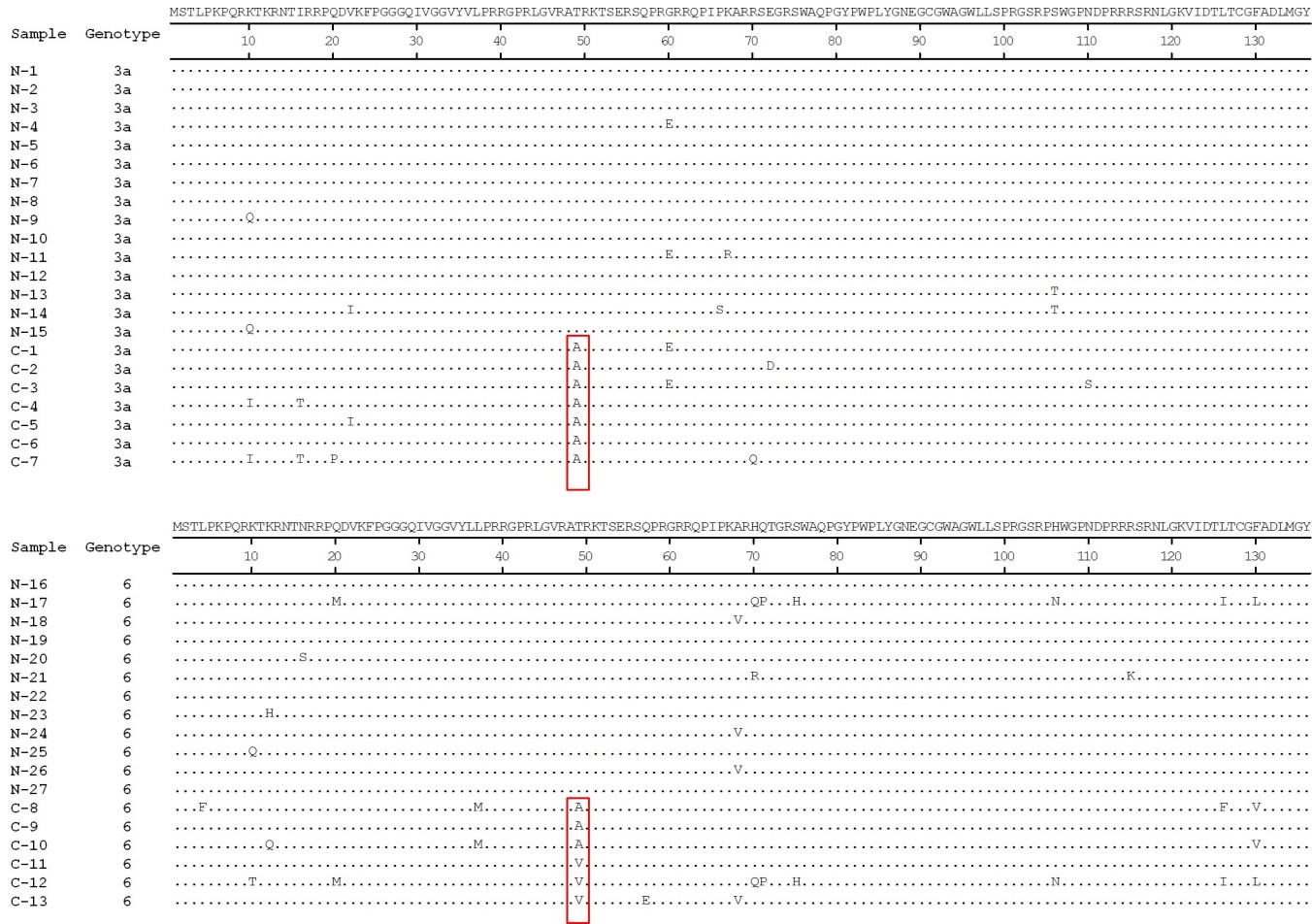

**Fig 2. Alignment of the partial amino acid sequences of HCVcAg belonging to genotype 3a and genotype 6.** Residue comparison of seven genotypes 3a samples of relatively low HCVcAg (C-1 to C-7) to other genotype 3a samples (N-1 to N-15). Residue comparison of six genotypes 6 samples of relatively low HCVcAg (C-8 to C-13) and other genotypes 6 samples (N-16 to N-27). Dots indicate identical residues. Boxes denote the threonine to alanine/valine change at position 49 observed for samples C-1 to C-13.

proteins. The scores of the 3D structure from the control sequence, the T49A mutant, and the T49V mutant were 0.3645, 0.3529, and 0.3630, respectively. These scores may be low because there is no available native structure of the full-length HCV core protein in the protein data bank (PDB) that can compare the first 136 residues of predicted HCV proteins. To evaluate the effect of the core mutation at position 49 on the tertiary structure of the HCV core protein, the 3D structure of the mutants (T49A/V) was compared to the control sequence. The replacements of threonine (T), a hydrophilic and polar amino acid, with the non-polar and hydrophobic amino acids, alanine (A) or valine (V), at position 49 were likely to be buried within the protein structure and obscured by neighboring amino acids (Fig 3).

## Discussion

In this study, almost half of the samples in this study were of HCV genotype 6, which is consistent with a previous report indicating that the most prevalent genotype in Phetchabun was genotype 6, followed by genotypes 3 and 1 [19, 20]. The findings shown here revealed a positive correlation between HCV RNA and HCVcAg levels, confirming the results of prior studies

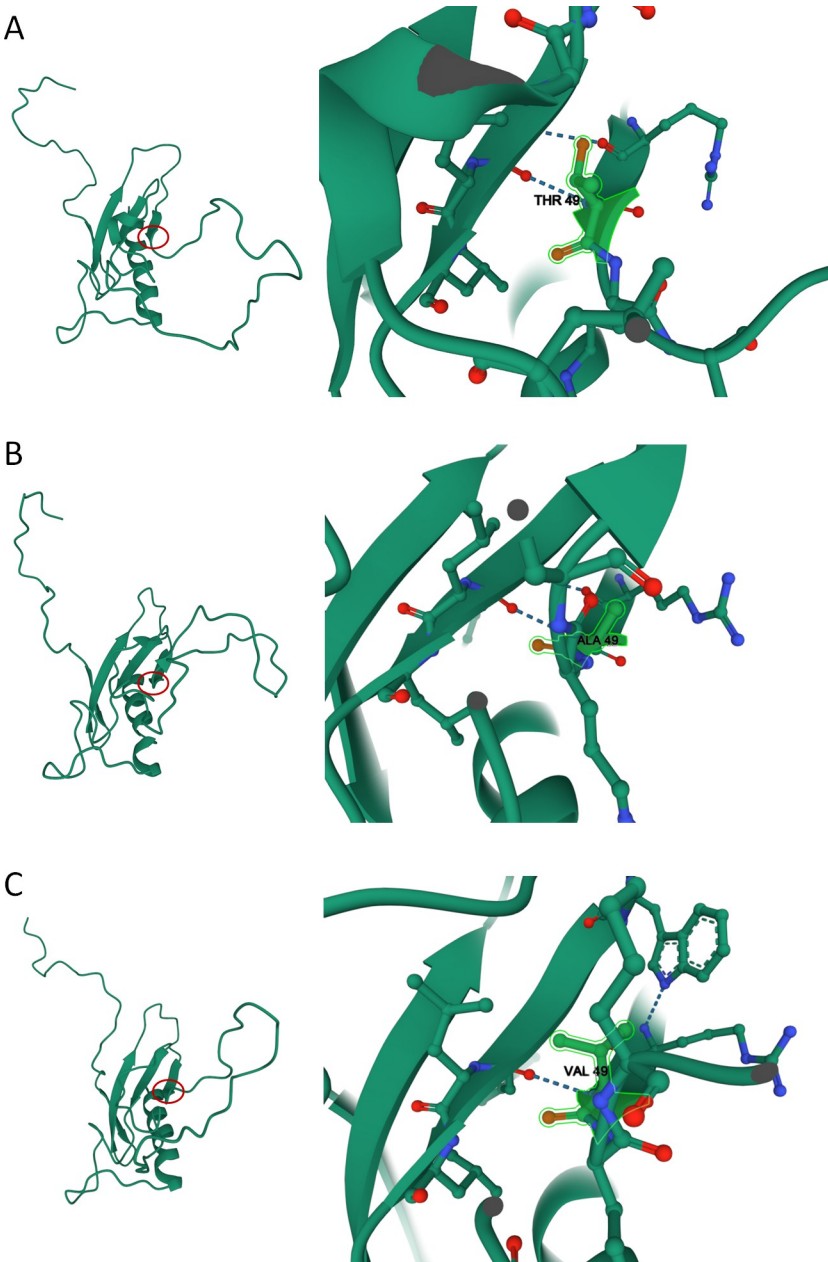

**Fig 3.** Tentative tertiary structure prediction of the HCV core protein from the control (A), T49A mutant (B), and T49V mutant (C) sequences. The global model quality score of the HCV core protein from the control sequence, T49A, and T49V mutant was 0.3645, 0.3529, and 0.3630, respectively.

[22, 23]. The correlation coefficients for all hepatitis C virus (HCV) genotypes demonstrated strong positive associations, ranging from 0.88 to 0.96 ($p < 0.001$). However, genotypes 3a and 6 exhibited weaker correlations compared to other genotypes, consistent with previous reports identifying greater variance in genotype 3a compared to genotypes 1a, 1b, and 4 [15]. Furthermore, our findings align with a previous study that detected greater variability of low HCVcAg levels and false negative results in samples from Thai patients with genotype 3a, potentially attributable to variations in amino acid mutations identified in this study [24].

The HCV core amino acid sequence is highly conserved among genotypes, particularly at positions 21–60 [25]. Mutation of the amino acid sequences in conserved regions may affect the sensitivity of core antigen detection [26]. Alanine and threonine at positions 48 and 49, respectively, are the most common residues in the core protein of all HCV genotypes. Substitution at these residues can occur in 0.1–15.7% of samples [18]. The current study found that 6% of HCV genotype 3a isolates had an amino acid substitution at position 49 (T49A). This matches the results from a previous study, which revealed a point mutation (T49P) in four (4%) of 107 HCV genotype 1b samples with lower HCVcAg relative to HCV RNA levels than control samples that lacked residue substitution [27]. In addition to HCV genotype 3a, a core mutation at position 49 (T49A/V) was also found in 4% of HCV genotype 6 isolates, the most prevalent genotype in Phetchabun [20].

Bioinformatics analysis predicts that the amino acid at position 49 of the HCV core region is part of the antibody epitope [28]. Mutation at residue 49 (T49P) reduced fluorescence enzyme immunoassay sensitivity [27]. Thus, core mutation at this position may correlate with one of the epitope regions recognized by an anti-HCV monoclonal antibody. Polar amino acids are usually found at the surface of proteins, while hydrophobic residues are mostly buried in the protein structure. In the current study, amino acid substitution at position 49 was accomplished by replacing polar and hydrophilic threonine with non-polar and hydrophobic alanine/valine. This substitution may affect the polarity, antigenicity, and tertiary structure of the HCV core protein, reducing the affinity of anti-HCV monoclonal antibody binding to HCVcAg. Multiple amino acid substitutions in the HCV core region (positions 47 to 49) may severely impair antibody binding, limiting the detection of HCVcAg [29]. Thus, HCVcAg levels may not be a reliable indicator of infection, especially among patients with low or undetectable levels. Inaccurate detection of HCVcAg can also lead to the wrong choice of treatment.

Three-dimensional modeling of the HCV core protein provides a tentative prediction of its tertiary structure. Tertiary structures were built directly from the amino acid sequence. No native structures of the full-length HCV core protein were available in the protein data bank (PDB) to compare with the predicted 3D structures. Thus, it remained unclear whether the predicted 3D structures had the correct fold and were close to the native structure. Moreover, the HCV core protein binding site for the anti-HCV monoclonal antibody used by the ARCHITECT HCV Ag platform could not be defined because the antibody molecule lacked structure in the protein-ligand interaction.

Regarding the potential use of HCVcAg as an alternative marker for detecting HCV infection, antigen assay has shown promising results for a diagnostic evaluation with high sensitivity, specificity, and accuracy (approximately 99%–100%) when tested on samples from HCV-infected Thai individuals [24]. However, the most significant variation in the correlation between HCV RNA and HCVcAg has been observed in genotype 3 samples, resulting in a 97% negative predictive value (2/290 samples were defined as negative by antigen testing). Our findings suggest that this may be associated with the T49A/V substitution, which is prevalent in approximately 6% of individuals with HCV genotype 3a in Thailand. If the antigen assay is used as a replacement for nucleic acid testing in a community screening, there is a risk of missing some HCV chronic patients who carry this mutation. Moreover, while the Thai NHSO currently implements HCVcAg as a confirmatory test for HCV diagnosis, policymakers should be aware of the assay's limitations and foster further follow-up studies to evaluate its utility, define the false negative rate, and determine the proportion of Thai patients who carry this mutation.

In summary, this study revealed a strong correlation between HCV RNA and HCVcAg levels. However, some samples had HCVcAg levels that were lower than those expected from the corresponding HCV RNA levels. These samples were classified as genotypes 3a and 6, which

are predominant in Phetchabun. In these samples, the amino acid alignment of the HCV core region had a point mutation at position 49 (T49A/V). Thus, while HCVcAg measurement may be used to identify infection when HCV RNA testing is unavailable, HCV RNA should remain the standard method for detecting active HCV infection and monitoring treatment.

## Supporting information

**S1 Fig. Phylogenetic analysis of HCV genotype 1a expansion.**
(TIF)

**S2 Fig. Phylogenetic analysis of HCV genotype 3a expansion.**
(TIF)

**S3 Fig. Phylogenetic analysis of HCV genotype 6f expansion.**
(TIF)

**S1 File. List of Core randomized samples.** The accession number of case (C) and control (N) of HCV core sample.
(DOCX)

## Acknowledgments

We are grateful to all staff from the sub-district health-promoting centers, district hospitals, and general hospitals in Phetchabun province. Without their collaboration, this research study may not have been possible.

## Author Contributions

**Conceptualization:** Payuda Hansoongnern, Yong Poovorawan.

**Data curation:** Rujipat Wasitthankasem, Nawarat Posuwan, Napaporn Pimsing.

**Formal analysis:** Payuda Hansoongnern, Prachya Kongtawelert.

**Funding acquisition:** Yong Poovorawan.

**Investigation:** Rujipat Wasitthankasem, Napaporn Pimsing, Yong Poovorawan.

**Methodology:** Payuda Hansoongnern, Pornpitra Pratedrat, Pornjarim Nilyanimit, Rujipat Wasitthankasem, Kanchanok Kodchakorn, Prachya Kongtawelert, Yong Poovorawan.

**Project administration:** Yong Poovorawan.

**Resources:** Payuda Hansoongnern, Yong Poovorawan.

**Validation:** Yong Poovorawan.

**Visualization:** Rujipat Wasitthankasem, Nasamon Wanlapakorn.

**Writing – original draft:** Payuda Hansoongnern, Nasamon Wanlapakorn, Yong Poovorawan.

**Writing – review & editing:** Payuda Hansoongnern, Rujipat Wasitthankasem.

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
