## [Decision Letter · Decision Letter 0]

22 Mar 2023

PONE-D-23-03115An amino acid substitution in HCV core antigen limits its use as a reliable measure of HCV infection compared with HCV RNAPLOS ONE

Dear Dr. Poovorawan,

Thank you for submitting your manuscript to PLOS ONE. After careful consideration, we feel that it has merit but does not fully meet PLOS ONE’s publication criteria as it currently stands. Therefore, we invite you to submit a revised version of the manuscript that addresses the points raised during the review process.

Please submit your revised manuscript by May 06 2023 11:59PM. If you will need more time than this to complete your revisions, please reply to this message or contact the journal office at plosone@plos.org. Please include the following items when submitting your revised manuscript:A rebuttal letter that responds to each point raised by the academic editor and reviewer(s). You should upload this letter as a separate file labeled 'Response to Reviewers'.A marked-up copy of your manuscript that highlights changes made to the original version. You should upload this as a separate file labeled 'Revised Manuscript with Track Changes'.An unmarked version of your revised paper without tracked changes. You should upload this as a separate file labeled 'Manuscript'.

We look forward to receiving your revised manuscript.

Kind regards,

Maemu Petronella Gededzha, Ph.D

Academic Editor

PLOS ONE

Journal Requirements:

"This study was part of a viral hepatitis elimination project in Thailand and was supported by the National Research Council of Thailand, Thailand Grand Challenge Fund (RES_64_058_30_020), Center of Excellence in Clinical Virology, Faculty of Medicine, Chulalongkorn University, and King Chulalongkorn Memorial Hospital. We are grateful to all staff from the sub-district health-promoting centers, district hospitals, and general hospitals in Phetchabun province. Without their collaboration, this research study may not have been possible. "

"NO - Include this sentence at the end of your statement: The funders had no role in study design, data collection and analysis, decision to publish, or preparation of the manuscript."

Additional Editor Comments:

When preparing your revised manuscript, you are asked to carefully consider the reviewer comments which are attached

Major:

i. It is not clear whether the samples used in this study are part of the samples tested positive in their previous study (Reference 13) or it’s a new cohort.

ii. If it is a new cohort please describe the cohort in details and indicate how many samples were screened by Anti-HCV?

iii. What is the sensitivity and specificity of the HCVcAg in this cohort?

iv. Include phylogenetic tree and indicate which method was used to construct the tree

v. Figure 1 resolution is poor and it is difficult to interpret the results

vi. Elaborate further in the discussion about the mutations detected in the core and treatment choice.

Minor comments:

vii. Please list the names, sources of all reagents used in the methodology.

viii. Reference the methods used eg. Sanger sequencing.

ix. Line 81-84-Write minutes and seconds in full.

x. Line 94: Which instrument was used? Eg i2000.

xi. Line 96-97: Where the specimens with concentration values ≥ 3.00 fmol/L to < 10.00 fmol/L retested in duplicate? What was the outcome?

xii. Line 106: Include the viral of the samples not detected.

Reviewers' comments:

Reviewer's Responses to Questions

**Comments to the Author**

1. Is the manuscript technically sound, and do the data support the conclusions?

Reviewer #1: Yes

Reviewer #2: Yes

2. Has the statistical analysis been performed appropriately and rigorously? 

Reviewer #1: Yes

Reviewer #2: Yes

3. Have the authors made all data underlying the findings in their manuscript fully available?

Reviewer #1: Yes

Reviewer #2: Yes

4. Is the manuscript presented in an intelligible fashion and written in standard English?

Reviewer #1: Yes

Reviewer #2: No

5. Review Comments to the Author

Reviewer #1: This is an absolutely important piece of data for HCV detection and eradication. The research cautions against the use of HCVcAg quantification as the sole method to test for HCV infection as some HCV genotypes with low levels of HCVcAg might results in false-negative detection of HCV.

A few comments to take note of and rectify or clarify:

Line 81 - mention the genotype of the reference strain.

Line 85 - Give an explanation for the use of two different amplification methods.

Line 89 - Accession numbers not available yet on GeneBank.

Line 93 - State the upper value first followed by the lower limit.

Line 111- Table 1 - add a column with the average HCV RNA levels.

Line 120 - 123 - Rephrase and avoid long sentences for clarity.

Line 150-151 - Rectify the sentence construction - plural.

Reviewer #2: The authors provide in this interesting work several evidences supporting the substitution of amino acid of HCV core antigen and the fact that it limit its use as reliable measure of HCV infection. They further showed that HCV RNA can be an alternative marker to be used. However there are several aspects to be improved in order to present convincing evidences and conclusions.

Major points

1. The introduction does not provide sufficient background of the study, it is therefor important that the author must write a comprehensive introduction.

2. The abstract needs some improvement

Minor points

3. Page 5, briefly explain the ARCHITECT HCV Ag assay

4. Page 6, line 117-18 need to be re-visited, the statement does not make sense at all. The explanation for the results need to be re-visited, especially on page 6, 7, it does not make sense at all. Results must be self-explanatory

5. It is important to discuss the method used to do phylogenetic analysis obtained

6. PLOS authors have the option to publish the peer review history of their article (what does this mean?). If published, this will include your full peer review and any attached files.

Reviewer #1: **Yes: **Hazel Tumelo Mufhandu

Reviewer #2: No

---

## [Author Response · Author response to Decision Letter 0]

1 May 2023

Response to reviewers

Editor Comments:

Question: It is not clear whether the samples used in this study are part of the samples tested positive in their previous study (Reference 13) or it’s a new cohort.

Answer: All samples used in this study were leftover from a previous population study, Pratedrat P; PlosOne 2023: PMID: 36656832 (reference 19). We have revised this section in material and method in lines 85-90. 

Question: If it is a new cohort, please describe the cohort in details and indicate how many samples were screened by Anti-HCV?

Answer: Samples in this study are part of a new cohort (reference 19). Briefly, in a total of 170,163 residents in Phetchabun province, Thailand, 10,621 tested positive for anti-HCV RDT. Among the HCV serological positive, 3,930 samples were positive for qualitative HCV RNA testing. Of these, 1,027 available samples were further tested for the viral load to fulfill the Thai Universal Health Coverage eligibility criteria for HCV treatment. A total of 354 serum samples in this study were the leftover samples from the 1027 samples. We have cited the new cohort reference in material and method (lines 85-90).

Question: What is the sensitivity and specificity of the HCVcAg in this cohort?

Answer: The manufacturer's instructions indicated that the sensitivity and specificity of the HCVcAg assay at a 3 fmol/L cut-off were 97.8% and 99.5%, respectively (lines 148-152). While the sensitivity and specificity of HCVcAg were not determined in the current cohort, a previous study on Thai HCV patients reported that the HCVcAg assay had a diagnostic performance of 99% sensitivity, 100% specificity, 100% positive predictive value, 97% negative predictive value, and 99% accuracy in predicting HCV active infection (Wasitthankasem R; Peer J 2017: PMID: 29134150). This information was discussed in lines 254-262.

Question: Include phylogenetic tree and indicate which method was used to construct the tree.

Answer: The HCV genotyping was classified by phylogenetic analysis using the Neighbor-joining approach with Kimura’s 2 parameters model (lines 121-133). The phylogenetic trees are included in the supplement figures. 

Question: Figure 1 resolution is poor and it is difficult to interpret the results.

Answer: We have improved the resolution accordingly.

Question: Elaborate further in the discussion about the mutations detected in the core and treatment choice.

Answer: We have discussed this point accordingly in lines 254-267.

Question: Please list the names and sources of all reagents used in the methodology.

Answer: We have included information about the reagents utilized in the Materials and Methods section, in lines 111-116.

Question: Reference the methods used eg. Sanger sequencing.

Answer: The information about the methods has been included in the Materials and Methods section, especially in lines 115-116 and 121-133.

Question: Lines 81-84, write minutes and seconds in full.

Answer: We have edited as a suggestion (lines 112-115).

Question: Line 94: Which instrument was used? Eg i2000.

Answer: The instrument used in this study is the Abbott Architect i1000SR analyzer (line 138) 

Question: Line 96-97: Where the specimens with concentration values ≥ 3.00 fmol/L to < 10.00 fmol/L retested in duplicate? What was the outcome?

Answer: To ensure accuracy, samples with a concentration ranging from ≥ 3.00 fmol/L to < 10.00 fmol/L underwent duplicate testing. All retested samples were found to be ≥ 3.00 fmol/L and therefore were assigned as reactive for HCVcAg (lines 148-150).

Question: Line 106: Include the viral of the samples not detected.

Answer: The results for two samples with HCV genotype 6 and HCV RNA levels of 4.04 x 103 and 8.85 x 103 IU/ml have been included in lines 162-163, indicating that despite their high RNA levels, they tested non-reactive for HCVcAg with values below 3 fmol/L.

Reviewer #1

This is an absolutely important piece of data for HCV detection and eradication. The research cautions against the use of HCVcAg quantification as the sole method to test for HCV infection as some HCV genotypes with low levels of HCVcAg might results in false-negative detection of HCV. A few comments to take note of and rectify or clarify:

Question: Line 81 - mention the genotype of the reference strain.

Answer: The primer positions were assigned according to the HCV prototype reference, H77, with accession number NC_004102 (lines 108-110).

Question: Line 85 - Give an explanation for the use of two different amplification methods.

Answer: The initial 76 samples were genotyped using primer sets that were too short of analyzing the 3D structure of the HCV core amino acid sequence. Furthermore, these samples could not be re-amplified for further PCR analysis. As a result, new primer sets were utilized for PCR amplification in the remaining 268 samples, which allowed for the analysis of a more extended nucleotide sequence and the 3D structure of the HCV core region (lines 100-105).

Question: Line 89 - Accession numbers not available yet on GeneBank.

Answer: The nucleotide sequences generated from this study were deposited in the GenBank database under accession numbers OQ351363-OQ351716. As per the GenBank policy, the sequence data will be made publicly available one year after submission to the GenBank repository or upon manuscript publication.

Question: Line 93 - State the upper value first, followed by the lower limit.

Answer: We have edited accordingly in line 137.

Question: Line 111- Table 1 - add a column with the average HCV RNA levels.

Answer: The average HCV RNA levels have been added to Table 1. 

Question: Line 120 - 123 - Rephrase and avoid long sentences for clarity.

Answer: The sentence has been revised accordingly (lines 173-178). 

Question: Line 150-151 - Rectify the sentence construction - plural.

Answer: The sentence has been revised accordingly in lines 204-207. 

Reviewer #2

The authors provide in this interesting work several evidences supporting the substitution of amino acid of HCV core antigen and the fact that it limits its use as reliable measure of HCV infection. They further showed that HCV RNA can be an alternative marker to be used. However, there are several aspects to be improved in order to present convincing evidences and conclusions.

Question: The introduction does not provide sufficient background of the study. It is therefore important that the author must write a comprehensive introduction.

Answer: The background knowledge has been included in more detail in the introduction (lines 50-62 and 67-73).

Question: The abstract needs some improvement

Answer: The abstract has been improved accordingly.

Question: Page 5, briefly explain the ARCHITECT HCV Ag assay

Answer: The ARCHITECT HCV Ag assay details were described in lines 141-152. 

Question: Page 6, line 117-18 need to be re-visited, the statement does not make sense at all. The explanation for the results needs to be re-visited, especially on page 6, 7, it does not make sense at all. Results must be self-explanatory.

Answer: The sentence has been revised accordingly (lines 173-178). 

Question: It is important to discuss the method used to do phylogenetic analysis obtained.

Answer: Phylogenetic trees were constructed using the Neighbor-joining approach using Kimura’s 2 parameters with 1000 bootstrap. The methodology of phylogenetic analysis used in this study was added in lines 121-133 and Figure S1-S3.

---

## [Decision Letter · Decision Letter 1]

12 Jun 2023

An amino acid substitution in HCV core antigen limits its use as a reliable measure of HCV infection compared with HCV RNA

PONE-D-23-03115R1

Dear Dr. Poovorawan,

We’re pleased to inform you that your manuscript has been judged scientifically suitable for publication and will be formally accepted for publication once it meets all outstanding technical requirements.

Kind regards,

Maemu Petronella Gededzha, Ph.D

Academic Editor

PLOS ONE

Additional Editor Comments (optional):

Reviewers' comments:

Reviewer's Responses to Questions

**Comments to the Author**

1. If the authors have adequately addressed your comments raised in a previous round of review and you feel that this manuscript is now acceptable for publication, you may indicate that here to bypass the “Comments to the Author” section, enter your conflict of interest statement in the “Confidential to Editor” section, and submit your "Accept" recommendation.

Reviewer #1: All comments have been addressed

Reviewer #2: All comments have been addressed

2. Is the manuscript technically sound, and do the data support the conclusions?

Reviewer #1: (No Response)

Reviewer #2: Yes

3. Has the statistical analysis been performed appropriately and rigorously? 

Reviewer #1: (No Response)

Reviewer #2: Yes

4. Have the authors made all data underlying the findings in their manuscript fully available?

Reviewer #1: (No Response)

Reviewer #2: Yes

5. Is the manuscript presented in an intelligible fashion and written in standard English?

Reviewer #1: (No Response)

Reviewer #2: Yes

6. Review Comments to the Author

Reviewer #1: (No Response)

Reviewer #2: (No Response)

7. PLOS authors have the option to publish the peer review history of their article (what does this mean?). If published, this will include your full peer review and any attached files.

Reviewer #1: **Yes: **Hazel Tumelo Mufhandu

Reviewer #2: **Yes: **Shonisani Wendy Limani

---

## [Editor Report · Acceptance letter]

20 Jun 2023

PONE-D-23-03115R1 

An amino acid substitution in HCV core antigen limits its use as a reliable measure of HCV infection compared with HCV RNA 

Dear Dr. Poovorawan:

I'm pleased to inform you that your manuscript has been deemed suitable for publication in PLOS ONE. Congratulations! Your manuscript is now with our production department. 

Kind regards, 

on behalf of

Dr. Maemu Petronella Gededzha 

Academic Editor

PLOS ONE